# An Investigation into the Potential of Turning Induced Deformation Technique for Developing Porous Magnesium and Mg-SiO_2_ Nanocomposite

**DOI:** 10.3390/ma16062463

**Published:** 2023-03-20

**Authors:** Michael Johanes, Manoj Gupta

**Affiliations:** Department of Mechanical Engineering, National University of Singapore, 9 Engineering Drive 1, Singapore 117575, Singapore; michael.johanes@u.nus.edu

**Keywords:** turning-induced deformation (TID), mechanical properties, microwave sintering, magnesium, porous materials, biodegradable implants

## Abstract

A new and novel method of synthesising porous Mg materials has been explored utilising a variant of a processing method previously used for the synthesis of dense Mg materials, namely the turning-induced deformation (TID) method combined with sintering. It was found that the Mg materials synthesised possessed comparable properties to previously-synthesised porous Mg materials in the literature while subsequent sintering resulted in a more consistent mechanical response, with microwave sintering showing the most promise. The materials were also found to possess mechanical response within the range of the human cancellous bone, and when reinforced with biocompatible silica nanoparticles, presented the most optimal combination of mechanical properties for potential use as biodegradable implants due to most similarity with cancellous bone properties.

## 1. Introduction

Magnesium as a metal has found widespread use in the engineering and biomedical sectors across many different applications. In particular, Mg-based alloys and composites have found substantial use as structural components in weight-critical applications such as aircraft parts and components [1]. However, they also show potential for use as medical implants in place of conventional materials such as titanium and steel [2]. These materials (Fe-, Ti- and Co-based) are dense and take the form of screws, plates, pins, etc. [3]. These materials over long-term usage show biological incompatibility due to the presence of alloying elements such as nickel, cobalt, and chromium [4].

In recent times, porous metallic materials have been studied for use as medical implants [5,6] or as energy-absorbing components [7,8,9]. These materials are most easily distinguishable from conventional dense materials by the presence of significant amounts of voids within the volume of the material, which may exceed 45% in some commercially available medical implants.

Porous Mg materials have been synthesized through various pathways, such as laser perforation [10,11], drilling [12], and the use of space holders [13,14,15,16]. The resulting porosity of these materials ranged from 19% to 50%, which can be compared against human cancellous bone possessing a porosity range of 50 to 90% [17,18,19]. Existing studies also reported the synthesis of porous Mg scaffolds by using space holder material [19] which is then removed and the resulting material subsequently sintered [11,20]. Coating of scaffolds helps to mitigate (lower) the initially high corrosion rate within application environments, yet at the same time use of these Mg scaffolds promotes better bone growth and recovery prospects [19,21,22]. Tissue ingrowth and implant-tissue integration was also improved with porosity and pore size and thus these can also serve to reduce the amount of screws or other implant fixtures [23], further increasing the attractiveness of porous materials for such purposes.

The literature study also indicated that these porous scaffolds can be processed for mechanical characterisation, but only after prior sintering or some other thermal/chemical bonding method [24,25,26] unless the scaffolds have already been synthesized/fabricated to dimensions immediately suitable for such mechanical tests [26,27].

A relatively new method for synthesising Mg materials is turning-induced deformation (TID), which involves severe plastic deformation of the Mg material through the turning process on a lathe machine [28]. Past works with this method entailed compaction of the aforementioned turnings, followed by extrusion of the compact into rods for further characterisation. It was further found that lower cutting speeds and higher depths of cut in the turning process resulted in superior mechanical response [29,30]. While past TID works focused on the synthesis of dense Mg materials, it is also feasible to synthesise porous Mg materials by utilising lower compaction pressures and omitting the subsequent extrusion step.

In terms of material composition, aside from the investigation of this method on pure Mg, Mg-2SiO_2_ nanocomposite was selected as it presented the best mechanical properties among Mg-SiO_2_ materials previously explored by Parande et al. [31]. In addition, SiO_2_ as a material also exhibits high biocompatibility [32,33], increasing its potential as an Mg material reinforcement in biomedical applications [34].

Thus, this work focuses, for the first time, on the application of TID in the synthesis of porous Mg materials for potential biomedical applications and engineering applications targeting energy absorption or impact such as in the transportation sector.

## 2. Materials and Methods

### 2.1. Materials

This work involved the synthesis of two different compositions using several raw materials; they are detailed in Table 1 and Table 2.

### 2.2. Synthesis

The raw materials were processed into ingots using the disintegrated melt deposition (DMD) method with a target superheat temperature of 750 °C, followed by stirring of the melt at 450 rpm for 5 min [35], after which turnings were generated from ingots using the TID secondary processing method [28] at 1.5 mm depth of cut and 55 mm/min average cutting speed. The resulting turnings were then compacted using a hydraulic press at a pressure of 1.03 MPa with a holding time of 60 s in cylindrical dies of 10 and 15 mm diameter to generate porous cylindrical compacts for further characterisation.

These samples were then characterized in as-compacted form, or additionally sintered using one of the following methods:Microwave sintering using a Sharp R-898C(S) 900W microwave for 18 min to reach 525 °C, with the setup cooling down inside the microwave chamber to ambient temperature.Furnace sintering using an Elite BAF7/15 furnace with a temperature of 520 °C at 20°/min temperature increase with a holding time of 2 h, with the setup cooling down without the lid until the Mg material compact reaches ambient temperature.

Schematics of the sintering setups are shown in Figure 1 and Figure 2.

The subsequent compacts are then designated according to Table 3.

### 2.3. Materials Characterisation

#### 2.3.1. Density and Porosity

Due to the nature and size of the samples (porous cylinders), the experimental density of the material was estimated by using the measured mass and dimensions assuming that the samples were perfect cylinders after grinding them flat for compression testing. The experimental porosity of individual samples used in the studies were then derived by comparing the experimental density to the theoretical material density.

#### 2.3.2. Microstructure

Samples were ground flat and finished using sandpaper of 4000 grit, followed by polishing using alumina suspension to 0.05-micron size using DI water as the lubricant to minimize surface oxidation. A Leica DM2500 optical microscope was used to obtain optical micrographs and a JEOL JSM-6010 scanning electron microscope equipped with energy dispersive X-ray (EDX) analysis capabilities was used to obtain scanning electron micrographs as well as material composition analysis results. MATLAB software (version R2013b) was also utilised to obtain average pore aspect ratios, pore diameter, and pore size distribution.

#### 2.3.3. X-ray Diffraction

A Shimadzu XRD-6000 X-ray diffractometer was used to conduct X-ray analysis of Mg and Mg-2SiO_2_ samples. The samples were exposed to Cu-Kα radiation (λ = 1.54056 Å) with a scanning speed of 2° per minute. The samples were scanned in the range of 10° to 70°. A graph of intensity (I) against 2θ (θ represents the Bragg angle) was obtained, and the observed peaks were compared against standard values of various phases.

#### 2.3.4. Grain Size

Flat and polished samples were etched using a solution of 4.2 g citric acid and 10 mL ethylene glycol in 100 mL of DI water, and grain morphology was quantified using MATLAB software (version R2013b) to obtain the average grain sizes of each material.

#### 2.3.5. Mechanical Properties

Microhardness characterization of the porous Mg samples was performed using a Shimadzu HMV-2 hardness tester with a test load of 245.2 mN at a dwell time of 15 s, with the top and bottom surfaces of samples measured and in accordance with ASTM E-384. A total of 15 measurements were taken across 1 surface or region of interest.

Flat and parallel samples with a length/diameter of 1 were subjected to compressive load testing using an MTS E-44 compressive tester machine according to procedures advised in standard ASTM E9-09, with a strain rate of 8.3 × 10^−5^ s^−1^ until failure. A minimum of 5 samples for each material and processing method combination were tested to obtain compressive properties.

## 3. Results and Discussion

### 3.1. Synthesis

The DMD ingot was free from cracks and turnings were generated without issues. Figure 3 shows photographs of TID turnings of the two materials, with the pure Mg turnings being more continuous than those of the Mg-2SiO_2_ material turnings which are both more discontinuous and powdery in comparison. SEM micrographs of these turnings in Figure 4 indicate that they, while distinct in appearance, retain the same features with one smooth side and a rougher side with shear bands corresponding to the induced plastic deformation on the machine turnings.

Figure 5 shows the resulting Mg material compacts from this processing method, clearly exhibiting the appearance and texture of the compacted turnings on the surfaces of the compacts.

### 3.2. Density and Porosity

The Mg-2SiO_2_ nanocomposite exhibited higher porosity than the pure Mg materials despite both having been synthesised using identical processing methods. The results are summarised in Table 4.

The porosity of the materials ranged from 10.7% to 17.2% in the case of the 15 mm die compacts and from 3.3% to 7.5% in the case of the 10 mm die compacts.

### 3.3. Microstructure

The surface morphology of the porous Mg material compacts clearly shows the presence of not only voids and pores, but also the morphology of the compacted turning surfaces. A notable trend was that the surface pores and voids closely follow the morphology of the compacted turnings, with less apparent pores and voids in sintered materials. Figure 6 below shows a comparison between unsintered and microwave-sintered TID Mg materials.

These pores were also measured for average aspect ratio and diameter, with Table 5 showing the results.

The aspect ratio of the pores generally increased with sintering, indicating a larger length-to-width ratio which is in line with the pores and voids being narrower. Further, the aspect ratio of pores was higher for 15 mm die compacts when compared to 10 mm die compacts. The presence of nano-SiO_2_ did not show any trend in changing the aspect ratio of pores. Pore diameter was affected the most in the case of 15 mm diameter pure Mg compacts, showing significant reductions in average diameter with sintering. The reverse trend was observed to a lesser degree in 10 mm diameter pure Mg and 15 mm diameter Mg-2SiO_2_ materials, whereby the average pore diameter increased. Figure 7 shows the normal distribution of pores by diameter, visualizing the effect of lower pore diameter standard deviations.

Microstructures of as-compacted and sintered Mg and Mg-2SiO_2_, showing the difference in morphology and appearance of spotted regions on the part of the latter, are shown in Figure 8. Figure 9 shows thinner spotted regions on furnace-sintered pure Mg.

For Mg-2SiO_2_ composites, EDX results display Mg, Si, and O in bulk materials and in chips confirming the presence of SiO_2_ nanoparticles (Figure 10 and Figure 11).

### 3.4. X-ray Diffraction

The resulting X-ray diffractograms of the TID Mg materials are shown in Figure 12.

Peaks corresponding to Mg, MgO, as well as Mg_2_Si were detected on all TID Mg materials regardless of sintering, while SiO_2_ was detected on TID Mg-2SiO_2_ materials.

### 3.5. Grain Size

The grain morphology differs significantly between as-compacted and sintered materials, as evident from Figure 13. The average grain sizes of the materials synthesised in this work are presented in Table 6. The sintered materials exhibited significantly larger grains than the fine-grained morphology of the as-compacted materials. The presence of nano-SiO2 reduced the average grain size of Mg in both as-compacted and sintered stages.

### 3.6. Mechanical Properties

#### 3.6.1. Microhardness Measurements

Microhardness values for the various materials are given in Table 7. For 15 mm compacts, the hardness remained the same (considering standard deviation) before and after sintering. However, for 10 mm compacts, sintering led to a reduction in average values both before and after sintering.

The Mg-2SiO_2_ material displayed higher hardness than that of pure Mg. Results also revealed that conventional furnace sintering resulted in higher hardness than microwave sintering.

#### 3.6.2. Compressive Properties

The resulting compressive properties are compiled in Table 8. For 15 mm die compacts, sintering reduced both the strengths and fracture strain. Conventional sintering showed an advantage over microwave sintering in terms of average values. The same observation was noted for 10 mm die compacts with regards to sintering except that strengths were better for microwave-sintered samples. The presence of SiO_2_ adversely affected the strength values both in as-compacted and sintered conditions.

Of the studied materials, the unsintered pure Mg exhibited superior compressive response, with Mg-15C-18M and Mg-2SiO_2_-15C-18M representing the best properties of the sintered materials. However, it was also found that sintered materials exhibit a more consistent compressive response than that of unsintered ones; this can be inferred from the lower absolute standard deviation values of the measured compressive properties despite roughly equal or inferior mean compressive response on the part of the sintered materials. This can also be seen from the actual compressive stress–strain curves of individual samples as seen in Figure 14 for Mg-2SiO_2_.

Table 9 compares the properties of TID porous Mg materials synthesised in this study against other porous Mg materials in previous works.

The features of the fractured samples following compression testing are shown in Figure 15 (macroscopic) and Figure 16 (microscopic). Fractography was conducted to gain insight into the fracture response of samples.

## 4. Discussion

### 4.1. Synthesis

Mg-2SiO_2_ and pure Mg were successfully synthesised using the TID method. The macrostructure results related to TID turnings are consistent with those found in previous TID works [28,29], and the compacts show that TID turnings were suitable for further processing into Mg compacts, though with random arrangements of turnings.

### 4.2. Density and Porosity

The higher porosity of 15 mm diameter compacts was due to the larger cavity within the compaction die, while being compacted with the same compaction pressure as that of the 10 mm diameter compacts. Mg-2SiO_2_ exhibited higher porosity than pure Mg, which can be attributed to the presence of the nanoparticles as opposed to the homogenous nature of pure Mg and the smaller chip size of Mg-2SiO_2_ material. It should also be noted that the experimental porosities of individual compacts studied in this work cannot be precisely controlled due to the random nature of the turnings used, with 18 min microwave-sintered compacts exhibiting the most consistency in porosity values as quantified by lower standard deviation values.

### 4.3. Microstructure

Spotted regions were observed in both TID Mg-2SiO_2_ materials as well as furnace-sintered Mg. For furnace-sintered Mg, these spotted regions were of thin, linear, amorphous structures while for Mg-SiO_2_, these regions resemble dendritic and lamellar structures containing Mg_2_Si much like those obtained by cast Al-Mg-Si alloys in other works [40,41]. The presence of Mg_2_Si is further confirmed by EDX results indicating the distributed presence of Si and O in addition to Mg across an area of Mg-2SiO_2_ turnings rather than just on a singular point as well as XRD results in Section 3.4.

The pore-narrowing effect of the sintering was first observed in this work, with the former being quantifiable with higher pore aspect ratios (narrower, less apparent pores and voids in line with visual microstructure observations), potentially reducing the degree of randomness of the pores contained within the Mg material compacts. This observation was further supported by the pore diameter results, where pore diameter standard deviations of sintered materials were generally lower. Gaining a more comprehensive understanding regarding this effect would require further analysis and investigation which are outside the scope of this study.

### 4.4. X-ray Diffraction

The observed peaks are in line with other work on sintered Mg-SiO_2_ materials, with Mg_2_Si being present in mechanically alloyed [42] and/or sintered materials [43] within the previous works. In addition, the SiO_2_ peak observed in Figure 12b was also in line with nanosized SiO_2_ in the literature [44]. The presence of both SiO_2_ and Mg_2_Si peaks on the unsintered TID Mg-2SiO_2_ materials thus indicates that the DMD process was sufficient to result in a partial reaction of SiO_2_ with Mg and the formation of Mg_2_Si within the material itself. Both SiO_2_ and Mg_2_Si peaks were still present after sintering indicating that the sintering was insufficient to completely react the remaining SiO_2_ with the Mg matrix, which was also observed in a previous work involving microwave sintering of powder-compacted Mg-2SiO_2_, whereby SiO_2_ nanoparticles were still visually present after sintering [31]. A singular, low-intensity Mg_2_Si peak was observed for furnace-sintered pure Mg. A possible explanation for this was the reaction of SiC with the Mg surface. This was previously reported in Mg-SiC composites synthesised using powder compaction, followed by sintering [45]. This would also explain the lack of Mg_2_Si peaks on the other TID pure Mg materials in this work since they were not in physical contact with SiC powder during synthesis.

### 4.5. Grain Size

The Hall–Petch relationship of increasing hardness with finer grain size has been observed [46]. In addition, a comparison can be drawn with the previously-mentioned prior study on Mg-2SiO_2_; the grain sizes of the pure Mg and Mg-2SiO_2_ TID materials in the present study are smaller than those generated without TID but with extrusion despite the grain refinement from dynamic recrystallisation [29,31,47,48] present in the former study.

### 4.6. Mechanical Properties

#### 4.6.1. Microhardness Measurements

The higher hardness of furnace-sintered Mg (relative to microwave-sintered Mg) and Mg-2SiO_2_ can be attributed to the well-distributed presence of SiO_2_ (for MgSiO2) and Mg_2_Si as observed with XRD, which possess hardness values far higher than that of pure Mg (7.3 MPa and 4.5 GPa for SiO_2_ and Mg_2_Si, respectively) [49,50]. Grain hardening achieved through lower grain sizes is in accordance with the Hall–Petch relationship [51]. Furthermore, smaller grains equate to larger grain boundary areas which have a larger probability of hindering dislocations at the grain boundary [52]. These factors all played a part in enhancing the hardness of Mg-2SiO_2_ beyond that of pure Mg.

A previous work by Parande et al. [31] involving powder metallurgy in generating extruded Mg-2SiO_2_ showed that the material microhardness was lower than that presented in the present study. This result showcases the superior properties imparted by the TID process despite forgoing the extrusion step which was performed in their study.

#### 4.6.2. Compressive Properties

The inferior compressive properties of furnace-sintered Mg (relative to microwave-sintered Mg materials) Mg-2SiO_2_ can be attributed to a higher amount of porosity and its pore aspect ratio (Table 4 and Table 5), as well as the presence of Mg_2_Si as was previously reported in Mg_2_Si-containing Mg composites [45].

The more consistent mechanical response of the sintered samples is significant as it represents more predictable porous material loading behaviour in actual conditions on a sample-by-sample basis, thereby narrowing the range of possible compressive response outcomes for a given material. This can be attributed to the previously-observed narrower pores with lower standard deviations obtained through sintering, resulting in less randomised pores with 18 min microwave sintering showing the most pronounced effect, as well as the material’s porosity having lower standard deviation. The sintered materials, though inferior in mechanical and/or compressive response are also closer to those of human cancellous bone in terms of mechanical properties, which is a beneficial characteristic as it minimises stress-shielding of bones as they heal [53,54].

From Table 7 above, it can be seen that the mechanical and compressive properties of TID porous Mg materials are comparable to those of previously-explored porous Mg materials except for lower porosity, and they also possess properties within the range of human cancellous bone with Mg-2SiO_2_-15C-18M being the closest of all materials explored in the present study.

The fractured pure Mg samples show a characteristic 45-degree angle of fracture, while Mg-2SiO_2_ materials exhibited crushing instead. Fractographs of both materials show that the fracture surfaces retain the original turning morphology, except those of microwave-sintered pure Mg which possess a resemblance to shear bands similar to dense Mg materials after a compressive fracture. This shows that microwave sintering was capable of providing additional bonding, if only for certain TID Mg materials.

## 5. Concluding Remarks and Future Work

Porous Mg materials of 3% to 17% were successfully synthesised using the TID method, followed by sintering. The following conclusions have also been observed:Porous TID Mg materials have shown comparable compressive response as well as other properties to other recent forms of porous Mg materials synthesised using different methods in the literature.Mg-2SiO_2_ shows potential as a better property-matching material with reference to human cancellous bone.Microwave sintering resulted in materials with the best consistency in mechanical response, and also enhances the compressive properties of porous TID Mg-2SiO_2_.

Thus, TID is viable as a potential processing pathway for porous Mg materials destined for biodegradable implant use from a mechanical standpoint. Further work in this field may involve the synthesis of more Mg materials using different parameters to further increase the porosity of the produced samples. This work would in turn result in a more comprehensive understanding of porous Mg materials fabricated using the TID method. Alternatively, different material compositions of porous Mg materials containing biocompatible reinforcements can also be studied to ascertain their mechanical response with the TID method.

## Figures and Tables

**Figure 1 materials-16-02463-f001:**
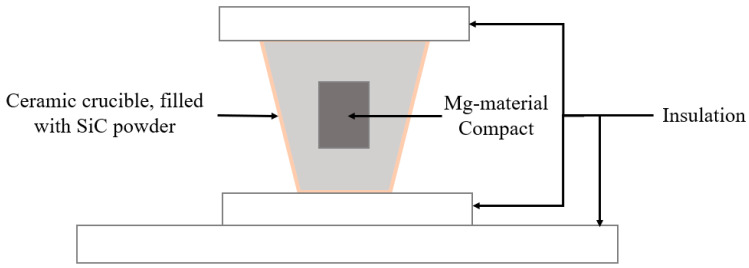
Schematic of furnace sintering setup of Mg material compacts.

**Figure 2 materials-16-02463-f002:**
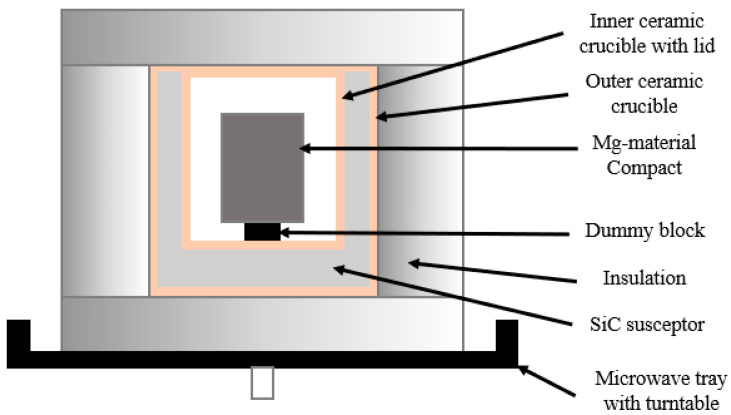
Schematic of microwave sintering setup of Mg material compacts.

**Figure 3 materials-16-02463-f003:**
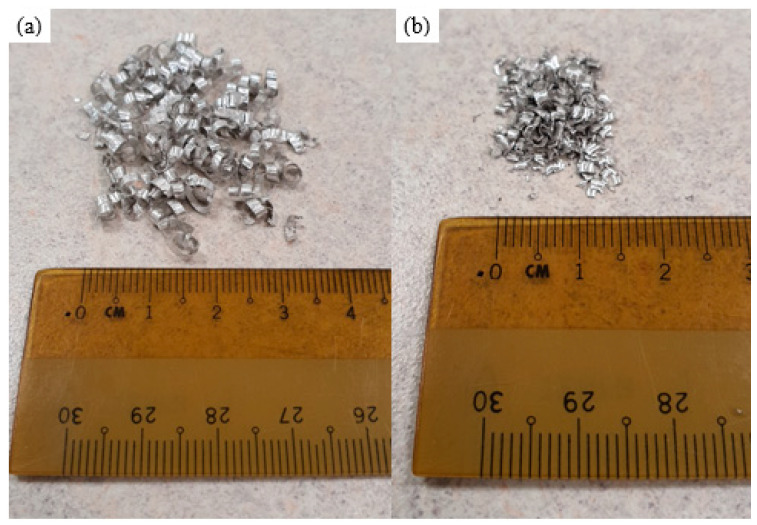
Turning photographs of (**a**) pure Mg and (**b**) Mg-2SiO_2_.

**Figure 4 materials-16-02463-f004:**
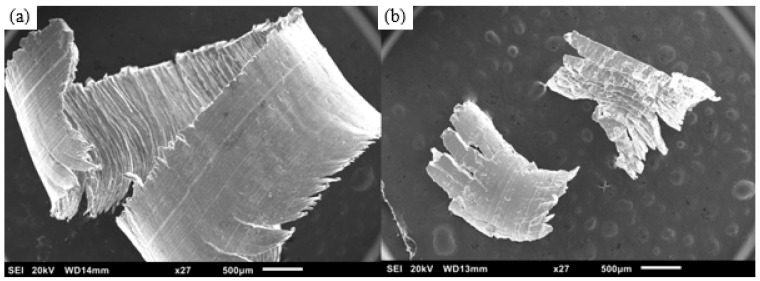
Scanning electron micrographs of turnings: (**a**) pure Mg and (**b**) Mg-2SiO_2_.

**Figure 5 materials-16-02463-f005:**
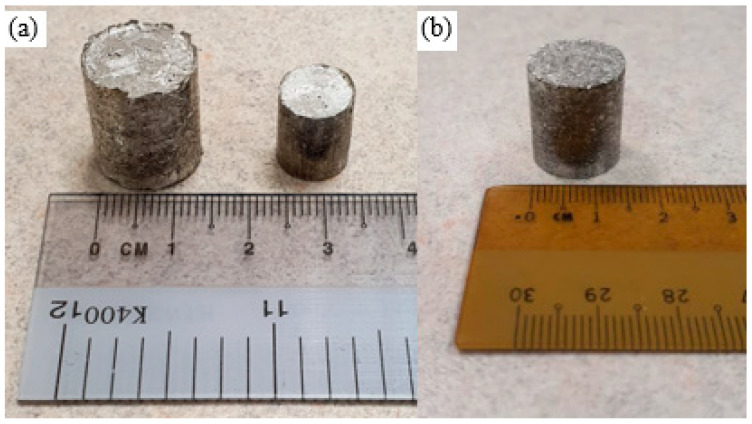
TID compacts of (**a**) pure Mg (left: 15 mm diameter, right: 10 mm diameter) and (**b**) Mg-2SiO_2_ (15 mm diameter).

**Figure 6 materials-16-02463-f006:**
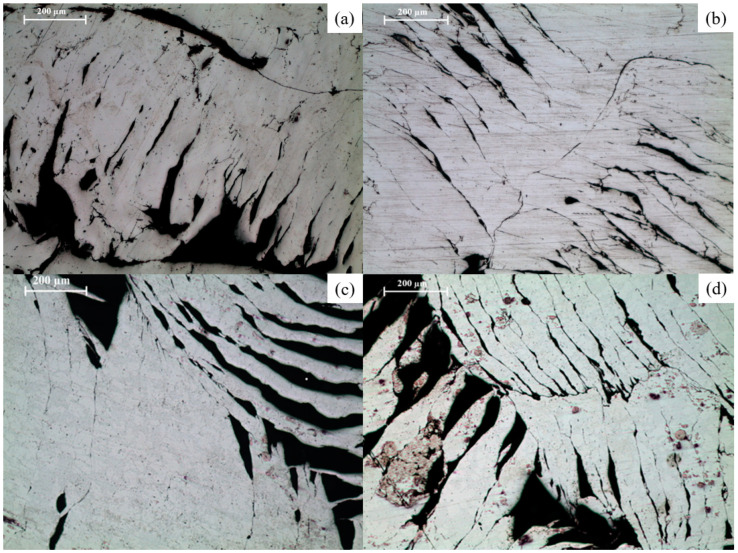
Optical micrographs of 15 mm compacts: (**a**) Mg-15C-N, (**b**) Mg-15C-18M, (**c**) Mg-2SiO_2_-15C-N, and (**d**) Mg-2SiO_2_-15C-18M.

**Figure 7 materials-16-02463-f007:**
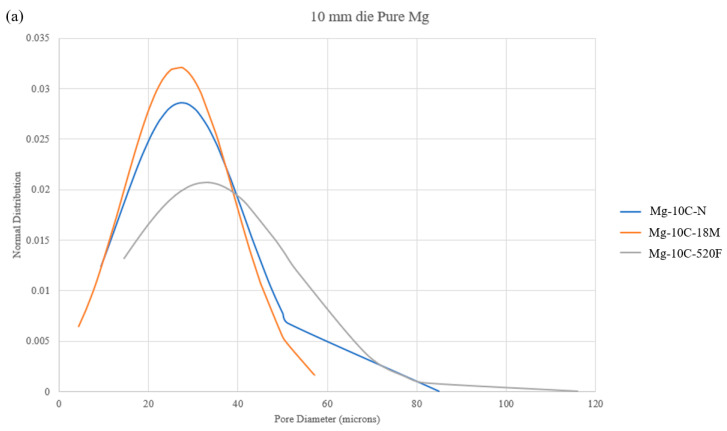
Normal distribution of pores by diameter for (**a**) 10 mm die pure Mg materials, (**b**) 15 mm die pure Mg materials, and (**c**) 15 mm die Mg-2SiO_2_ materials.

**Figure 8 materials-16-02463-f008:**
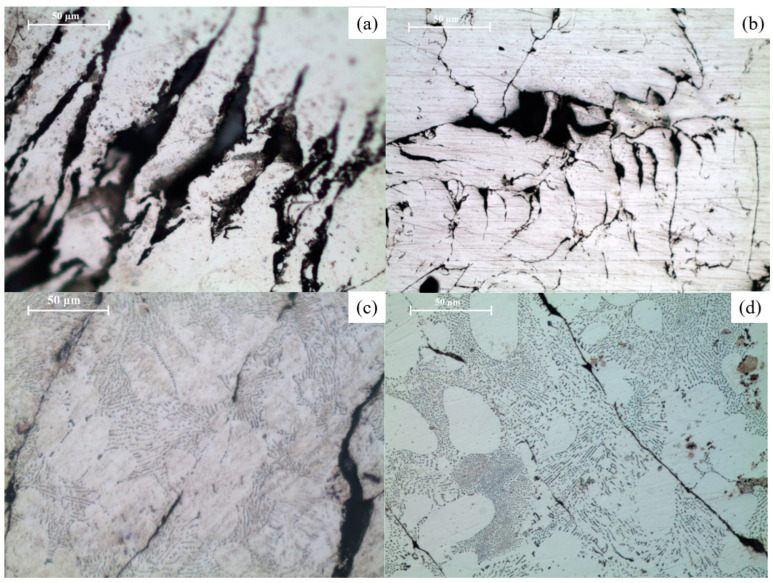
Optical micrographs of 15 mm compacts: (**a**) Mg-15C-N, (**b**) Mg-15C-18M, (**c**) Mg-2SiO2-15C-N, and (**d**) Mg-2SiO2-15C-18M.

**Figure 9 materials-16-02463-f009:**
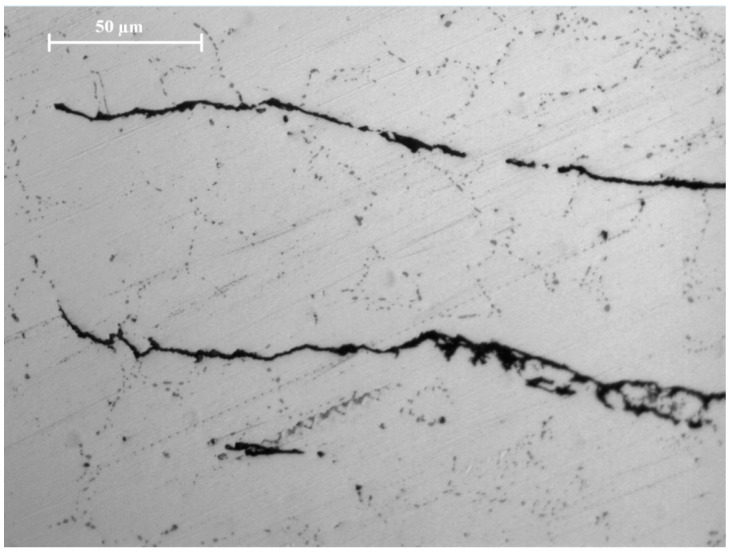
Optical micrograph of Mg-15C-520F.

**Figure 10 materials-16-02463-f010:**
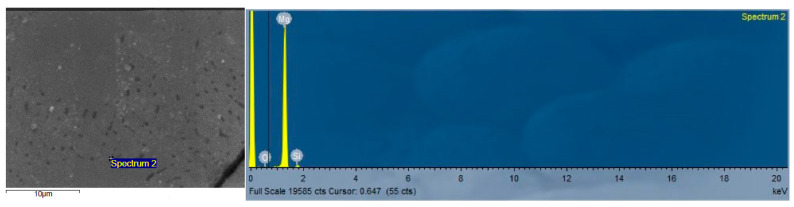
Scanning electron micrograph of selected EDS analysis location on TID Mg-2SiO_2_ spotted region (**left**) and the detected spectrum showing Mg, Si, and O peaks (**right**).

**Figure 11 materials-16-02463-f011:**
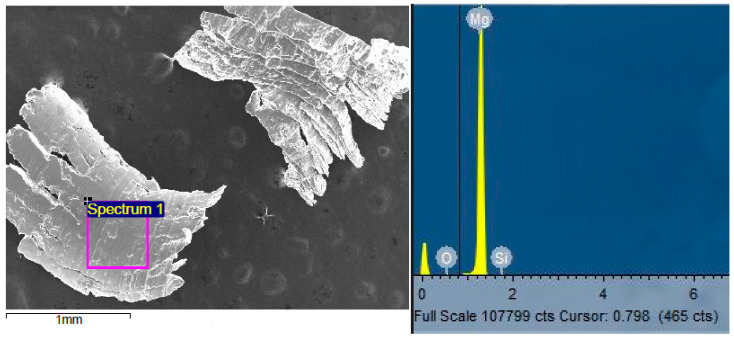
Scanning electron micrograph of selected EDS analysis location on Mg-2SiO_2_ turning (**left**) and the detected spectrum showing Mg, Si, and O peaks (**right**).

**Figure 12 materials-16-02463-f012:**
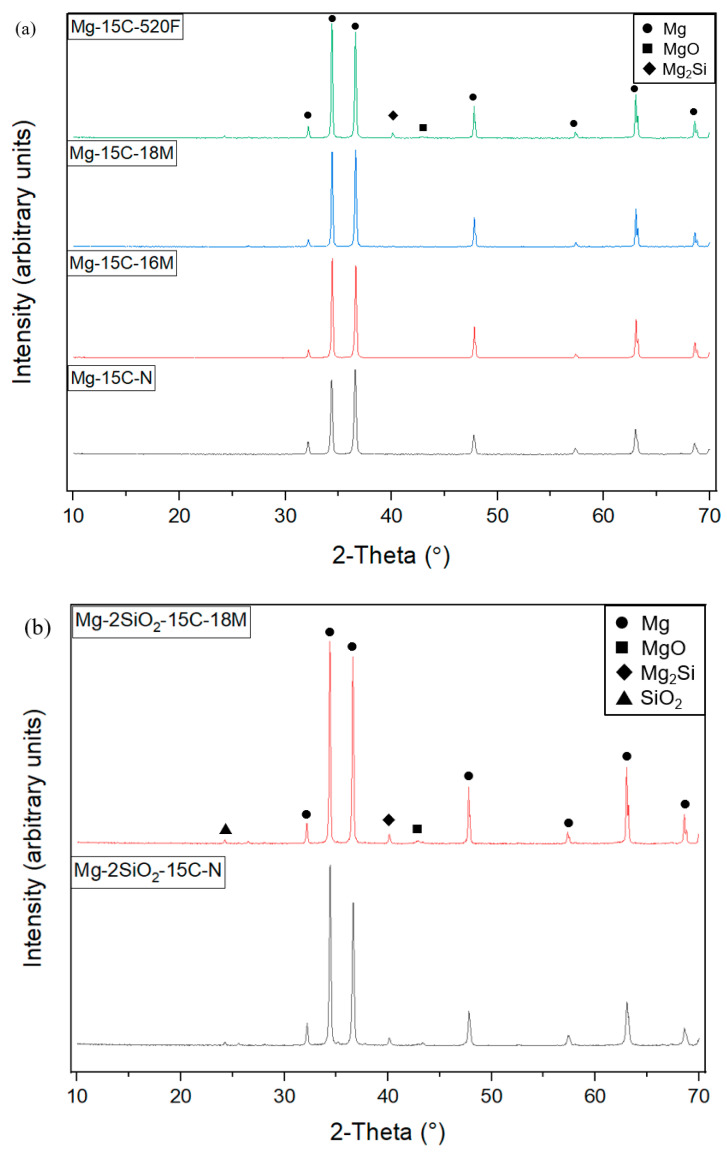
X-ray diffractograms of 15 mm TID compacts of (**a**) pure Mg, (**b**) Mg-2SiO_2_.

**Figure 13 materials-16-02463-f013:**
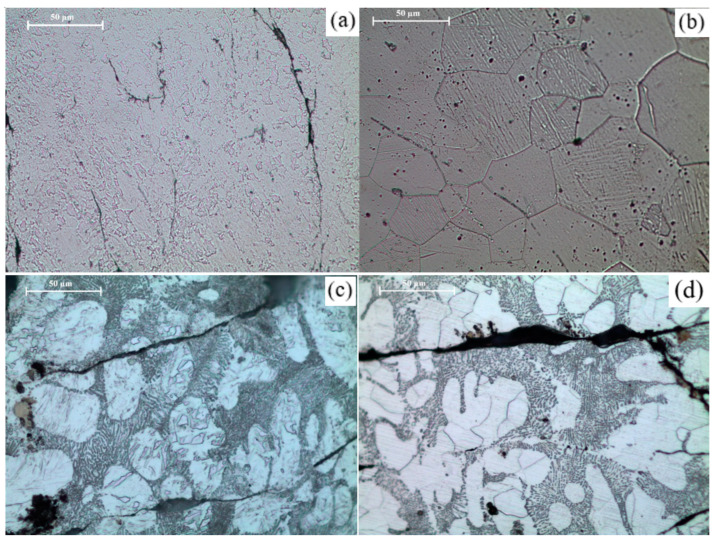
Optical micrographs of (**a**) Mg-15C-N, (**b**) Mg-15C-18M, (**c**) Mg-2SiO_2_-15C-N, and (**d**) Mg-2SiO_2_-15C-18M.

**Figure 14 materials-16-02463-f014:**
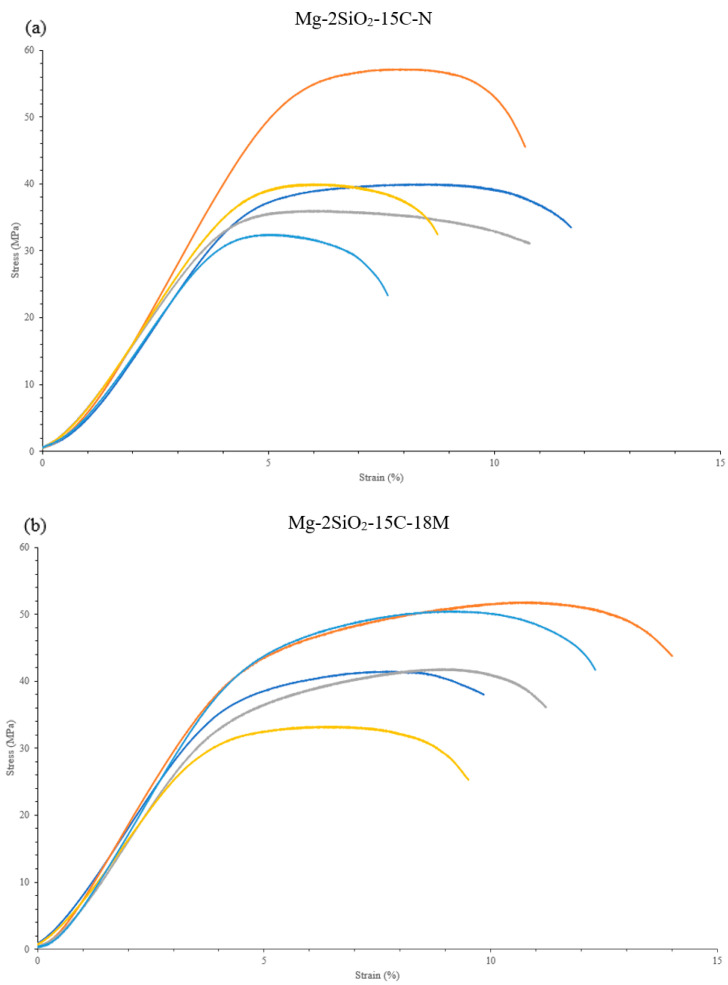
Compressive stress–strain curves of individual Mg-2SiO_2_ TID compacts for (**a**) as-compacted (**b**) 18 min microwave-sintered.

**Figure 15 materials-16-02463-f015:**
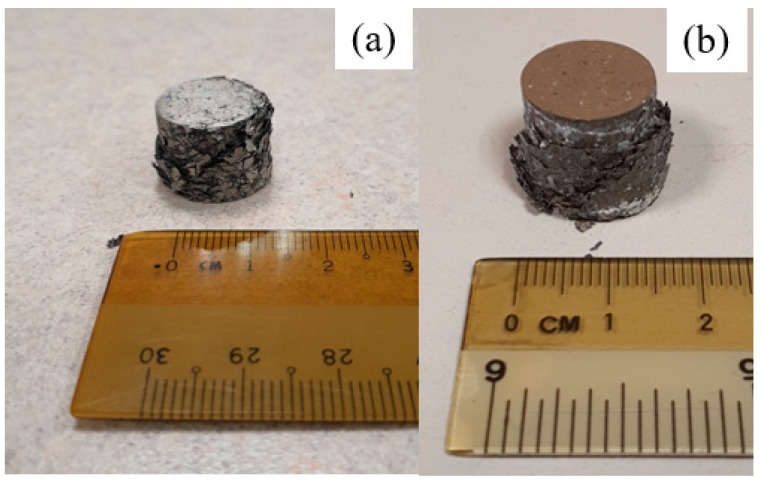
Fractured TID 15 mm compacts for (**a**) pure Mg and (**b**) Mg-2SiO_2_.

**Figure 16 materials-16-02463-f016:**
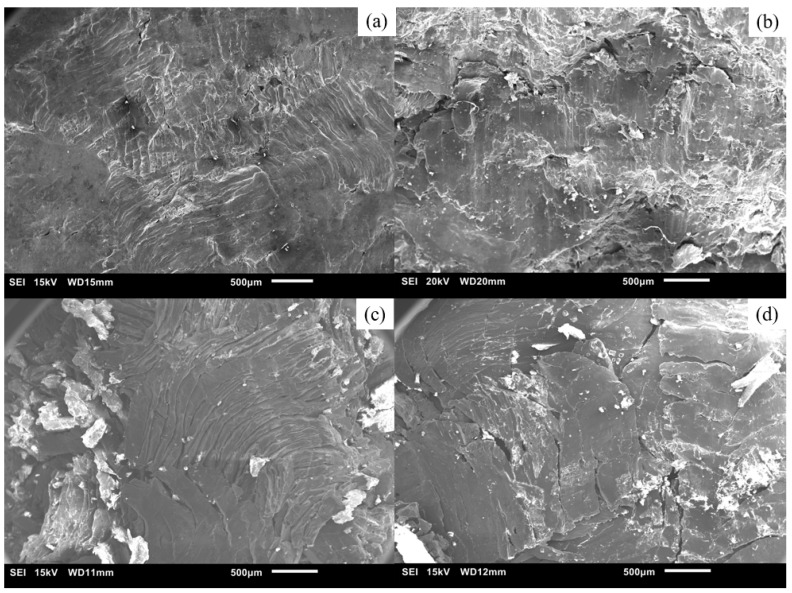
Scanning electron micrographs of TID 15 mm compact fracture surfaces with condition: (**a**) Mg-15C-N, (**b**) Mg-15C-18M, (**c**) Mg-2SiO_2_-15C-N, and (**d**) Mg-2SiO_2_-15C-18M.

**Table 1 materials-16-02463-t001:** Raw materials used in this work.

Raw Material	Supplier
Mg turnings, 99.9% purity	Acros Organics (Morris Plains, NJ, USA)
SiO_2_ nanopowder, 10–20 nm	Sigma Aldrich (Singapore)

**Table 2 materials-16-02463-t002:** Mg materials synthesised in this work, alongside their composition and theoretical densities derived from the rule of mixtures.

Material Composition	Raw Materials Composition by Weight%	Theoretical Density (g/cm^3^)
Pure Mg	100% Mg	1.738
Mg-2SiO_2_	98% Mg, 2% SiO_2_	1.748

**Table 3 materials-16-02463-t003:** Designation and processing of Mg materials synthesised in this work.

Designation	Processing Conditions	
	Compaction	Sintering
Mg-15C-N	15 mm die @ 1.03 MPa	N/A
Mg-15C-16M		16 min microwave sinter
Mg-15C-18M		18 min microwave sinter
Mg-15C-520F		Furnace sinter to 520 °C
Mg-2SiO_2_-15C-N		N/A
Mg-2SiO_2_-15C-18M		18 min microwave sinter
Mg-10C-N	10 mm die @ 1.03 MPa	N/A
Mg-10C-18M		18 min microwave sinter
Mg-10C-520F		Furnace sinter to 520 °C

**Table 4 materials-16-02463-t004:** Average porosities of Mg materials synthesised in this work.

Material	Average Porosity (%)
Mg-15C-N	10.7 ± 2.1
Mg-15C-16M	13.3 ± 2.2 ↑
Mg-15C-18M	12.9 ± 0.7 ↑
Mg-15C-520F	11.4 ± 2.0 ↑
Mg-2SiO_2_-15C-N	17.2 ± 2.1
Mg-2SiO_2_-15C-18M	13.9 ± 0.7 ↓
Mg-10C-N	7.5 ± 3.5
Mg-10C-18M	3.3 ± 0.6 ↓
Mg-10C-520F	6.7 ± 2.0 ↓

**Table 5 materials-16-02463-t005:** Average pore aspect ratios and diameters of Mg materials synthesised in this work.

Material	Average Pore Aspect Ratio	Average Pore Diameter (Microns)
Mg-15C-N	4.6 ± 1.9	59 ± 50
Mg-15C-16M	3.9 ± 1.9 ↓	32 ± 24 ↓
Mg-15C-18M	6.5 ± 2.2 ↑	33 ± 15 ↓
Mg-15C-520F	7.3 ± 3.8 ↑	34 ± 7 ↓
Mg-10C-N	2.0 ± 0.8	27 ± 14
Mg-10C-18M	4.8 ± 2.4 ↑	27 ± 12
Mg-10C-520F	3.4 ± 1.1 ↑	33 ± 19 ↑
Mg-2SiO_2_-15C-N	5.1 ± 2.5	98 ± 78
Mg-2SiO_2_-15C-18M	6.5 ± 2.7 ↑	110 ± 72 ↑

**Table 6 materials-16-02463-t006:** The average grain size of Mg materials synthesised in this work.

Material	Average Grain Diameter (Microns)
Mg-15C-N	8 ± 2
Mg-15C-16M	25 ± 11 ↑
Mg-15C-18M	28 ± 10 ↑
Mg-15C-520F	21 ± 7 ↑
Mg-10C-N	6 ± 2
Mg-10C-18M	22 ± 7 ↑
Mg-10C-520F	22 ± 7 ↑
Mg-2SiO_2_-15C-N	5 ± 2
Mg-2SiO_2_-15C-18M	16 ± 5 ↑
Mg-2SiO_2_ (extruded) [31]	23 ± 2

**Table 7 materials-16-02463-t007:** The average hardness of Mg materials synthesised in this work.

Material	Average Microhardness (H_v_)
Mg-15C-N	65 ± 5
Mg-15C-16M	61 ± 7 ↓
Mg-15C-18M	60 ± 8 ↓
Mg-15C-520F	65 ± 6 ↓
Mg-10C-N	74 ± 4
Mg-10C-18M	58 ± 6 ↓
Mg-10C-520F	62 ± 8 ↓
Mg-2SiO_2_-15C-N	83 ± 10
Mg-2SiO_2_-15C-18M	73 ± 9 ↓
Mg-2SiO_2_ (extruded) [31]	69 ± 2

**Table 8 materials-16-02463-t008:** Compressive properties of Mg materials synthesised in this work.

Material	Mean Young’s Modulus (GPa)	Mean 0.2% Yield Strength (MPa)	Mean Ultimate Compressive Strength (MPa)	Mean Fracture Strain (%)
Mg-15C-N	1.27 ± 0.08	52.69 ± 6.39	87.56 ± 15.76	32.78 ± 4.68
Mg-15C-16M	1.20 ± 0.10 ↓	34.91 ± 3.69 ↓	73.39 ± 8.59 ↓	27.03 ± 2.93 ↓
Mg-15C-18M	1.09 ± 0.03 ↓	30.73 ± 0.90 ↓	76.31 ± 2.00 ↓	25.25 ± 0.89 ↓
Mg-15C-520F	1.06 ± 0.16 ↓	35.74 ± 6.75 ↓	78.87 ± 7.00 ↓	22.52 ± 3.11 ↓
Mg-10C-N	1.49 ± 0.21	63.67 ± 13.92	84.43 ± 20.15	27.3 ± 6.00
Mg-10C-18M	1.46 ± 0.12 ↓	47.62 ± 4.34 ↓	108.93 ± 5.28 ↑	22.7 ± 1.12 ↓
Mg-10C-520F	1.07 ± 0.36 ↓	38.79 ± 5.51 ↓	89.54 ± 11.29 ↑	26.15 ± 3.40 ↓
Mg-2SiO_2_-15C-N	1.04 ± 0.09	36.25 ± 6.28	41.02 ± 8.51	9.91 ± 1.49
Mg-2SiO_2_-15C-18M	1.05 ± 0.07 ↑	33.66 ± 4.47 ↓	43.68 ± 6.77 ↑	11.38 ± 1.65 ↑
Cancellous bone [17,18,19,36,37]	0.1–20	-	2–48	-

**Table 9 materials-16-02463-t009:** Compressive properties of Mg materials synthesised in this work alongside other porous Mg materials.

Material	Average Porosity (%)	Mean Young’s Modulus (GPa)	Mean 0.2% Yield Strength (MPa)	Mean Ultimate Compressive Strength (MPa)	Mean Compressive Fracture Strain (%)
Mg-10C-18M (this study)	3.26	1.46 ± 0.12	47.62 ± 4.34	108.93 ± 5.28	22.7 ± 1.12
Mg-15C-18M (this study)	12.85	1.09 ± 0.03	30.73 ± 0.90	76.31 ± 2.00	25.25 ± 0.89
Mg-2SiO_2_-15C-18M (this study)	13.94	1.05 ± 0.07	33.66 ± 4.47	43.68 ± 6.77	11.38 ± 1.65
Porous Mg [12]	30–55	1.45–2.21	-	63–103	6.4–7.1
Porous Mg [13]	62–75	-	0.5–20	4–14	<5
Porous Mg [14]	29–31	0.6–1.2	13–53	20–70	5–17
Porous Mg [15]	50	0.35	-	6–7	<5
Cancellous bone [17,18,19,36,37,38,39]	50–90	0.1–-20	-	2–48	1.11–4.0

## Data Availability

Not applicable.

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
