# Peer review of "An Investigation into the Potential of Turning Induced Deformation Technique for Developing Porous Magnesium and Mg-SiO2 Nanocomposite"

_materials, 2023, doi:10.3390/ma16062463_

Round 1
Reviewer 1 Report
Overall, this manuscript is very strong and the research is sound. I found some grammatical and word choice improvements that could be made which are marked in the attached. Also, the legends in Figure 13 were missing and need to be added. It is also not clear which peak the authors are referring to that corresponds to silica in Figure 11b.

Reviewer 2 Report
This is a well-written and presented engineering paper.
My only criticism is that there is no report of mean pore sizes or pore size distribution. Different pore sizes are likely to be altered at different rates during sintering; for example, do small pore radii reduce more quickly than large pore radii?.
A clue to this possibility may perhaps be seen in Table 9, where the yield strength and compressive fracture strain do not vary linearly with average porosity.
